# Embedding stakeholder preferences in setting priorities for health research: Using a discrete choice experiment to develop a multi-criteria tool for evaluating research proposals

William J. Taylor[1,2,3]*, Haitham Tuffaha[4], Carmel M. Hawley[5,6,7], Philip Peyton[8], Alisa M. Higgins[9], Paul A. Scuffham[10], Fiona Nemeh[11], Anitha Balagurunathan[11], Paul Hansen[12], Angela Jacques[13], Rachael L. Morton[14]

1 Department of Medicine, University of Otago, Wellington, New Zealand, 2 Hutt Valley District Health Board, Lower Hutt, New Zealand, 3 Tairawhiti District Health Board, Gisborne, New Zealand, 4 Centre for the Business and Economics of Health, University of Queensland, Brisbane, Australia, 5 Australasian Kidney Trials Network (AKTN), Brisbane, Australia, 6 Department of Nephrology, Princess Alexandra Hospital, Brisbane, Australia, 7 Translational Research Institute, Brisbane, Queensland, Australia, 8 Australia and New Zealand College of Anaesthetists Clinical Trials Network, Melbourne, Australia, 9 Australia and New Zealand Intensive Care-Research Centre, Monash University, Melbourne, Australia, 10 Menzies Health Institute Queensland, Brisbane, Australia, 11 Australian Clinical Trials Alliance, Melbourne, Australia, 12 Department of Economics, University of Otago, Dunedin, New Zealand, 13 Institute for Health Research, The University of Notre Dame, Freemantle, Australia, 14 NHMRC Clinical Trials Centre, Sydney, Australia

* will.taylor@otago.ac.nz

**Data Availability Statement:** De-identified datasets are available at https://doi.org/10.6084/m9.

## Abstract

We determined weights for a multi-criteria tool for assessing the relative merits of clinical-trial research proposals, and investigated whether the weights vary across relevant stakeholder groups. A cross-sectional, adaptive discrete choice experiment using 1000minds online software was administered to consumers, researchers and funders affiliated with the Australian Clinical Trials Alliance (ACTA). We identified weights for four criteria—*Appropriateness*, *Significance*, *Relevance*, *Feasibility*—and their levels, representing their relative importance, so that research proposals can be scored between 0% (nil or very low merit) and 100% (very high merit). From 220 complete survey responses, the most important criterion was *Appropriateness* (adjusted for differences between stakeholder groups, mean weight 28.9%) and the least important was *Feasibility* (adjusted mean weight 19.5%). Consumers tended to weight *Relevance* more highly (2.7% points difference) and *Feasibility* less highly (3.1% points difference) than researchers. The research or grant writing experience of researchers or consumers was not associated with the weights. A multi-criteria tool for evaluating research proposals that reflects stakeholders' preferences was created. The tool can be used to assess the relative merits of clinical trial research proposals and rank them, to help identify the best proposals for funding.

figshare.21067888 and https://doi.org/10.6084/
m9.figshare.22580431 under a CC BT 4.0 licence.

**Funding:** The authors received no specific funding
for this work.

**Competing interests:** The authors have declared
that no competing interests exist.

## Introduction

Clinical trial networks (CTNs) are formalised collaborations of researchers and clinicians who make use of their expertise and economies of scale to leverage multi-centre randomised controlled trials (RCTs) that aim to address issues of clinical importance. Some CTNs are based on professional societies, whereas others are structured around particular conditions or therapies. Research is typically investigator-initiated and funded from non-commercial sources, such as universities, charities and governments. In order to use public funding prudently and allocate resources efficiently, CTNs must carefully choose which projects to pursue. A framework for prioritising the most worthwhile research would reduce the wastage associated with low-value published research identified by Glasziou and Chalmers [1] and others [2].

We previously identified three main approaches to research prioritisation: interpretive (e.g., James Lind Alliance [3]), quantitative (e.g., value of information analyses [4]) and blended approaches (e.g., multi-criteria decision analysis [MCDA] [5, 6]). We found that a blended approach has significant advantages, including transparency, ease of use and broad stakeholder involvement. However, existing scoring systems that quantify the relative merits of research proposals tend to use arbitrarily derived weights that are unlikely to reflect all stakeholders' views [7].

Informed by a previous literature review and stakeholder survey [8], and described in detail there, we developed a set of four criteria capturing the key factors that represent the 'worth' or merit of a research project. In brief, these four criteria are:

- *Appropriateness*—should the research be done? The extent to which the study design and study quality are appropriate to answer the research question.

- *Significance*—what does society get out of it? The extent to which society overall benefits or capacity, partnerships or innovation are enhanced.

- *Relevance*—why do the research? The extent to which high-burden conditions or stakeholder needs are addressed.

- *Feasibility*—can it be done? The extent to which researchers are able to deliver outputs on time and within budget.

Using this framework, supported by the Australian Clinical Trials Alliance (ACTA), we used a MCDA approach—specifically, a discrete choice experiment—to develop a multi-criteria tool for assessing the relative merits of clinical-trial research proposals. The aim was to create a tool that would help stakeholders better understand proposals' relative merits in broad terms. The MCDA approach involved specifying the criteria for assessing proposals, and their levels of performance, and determining weights for the criteria and levels that represent their relative importance to stakeholders. Such a tool is ultimately applied by rating each proposal on the criteria and summing the weights to get a total score, by which the proposals can be ranked.

## Methods

Participants in the study were invited to complete a survey based on an adaptive discrete choice experiment (DCE) [9] implemented using 1000minds software (www.1000minds.com). A DCE is a popular research methodology for identifying participants' stated preferences with respect to the relative importance of the criteria for prioritizing the alternatives of interest—in the present context, research proposals.

A major advantage of the DCE method (explained below) used by 1000minds to capture the relative importance of the criteria is that a set of weights is generated for each individual

DCE participant, in contrast to most other DCE methods that produce parameters of a model derived from aggregated data. Individual-level data enables the extent to which participants' socio-demographic and background characteristics explain patterns in their weights to be investigated.

The DCE survey was based on the four criteria discussed in the previous section—*Appropriateness*, *Significance*, *Relevance and Feasibility* -, with clarifying statements provided to help participants understand the criteria and their levels. The survey was pilot tested by 5 researchers and 3 consumers (defined as users of healthcare services).

For each criterion, the levels of performance on each criterion were defined in as broad and generic terms as possible: high, medium-high, medium, medium-low, low. Because five levels resulted in a relatively large number of questions being asked in the DCE—too many for most participants to answer comfortably—only the first, third and fifth levels (low, medium, high) appeared in the DCE. The values of the two intermediate levels (medium-high, medium-low) were interpolated by the software by fitting a monotonic smoothed curve through the weights for the other three levels.

It is worthwhile noting that how participants interpret these levels in the DCE is relative to their own personal experiences and not intended to be interpersonally comparable. For example, what one person thinks is a medium performance on any of the four criteria could be a low performance for another person or a high performance for another, but that is irrelevant here because the objective is to discover the criteria's relative importance to each individual. As mentioned above, an important strength of the PAPRIKA method is that a set of weights is generated for each individual.

1000minds implements the PAPRIKA DCE method—an acronym for "Potentially All Pairwise RanKings of all possible Alternatives" [10]. This software and method have been used in a wide range of health applications since 2004; for a recent short survey see Sullivan et al [11]. The PAPRIKA method involves each participant being asked a series of questions—known as 'choice tasks'—based on choosing between pairs of hypothetical research proposals characterised by just two criteria at a time and involving a trade-off, where implicitly the levels on all other criteria are the same for both proposals (see Fig 1 for an example).

These choice tasks, based on just two criteria at a time—known as 'partial profiles' (in contrast to 'full profiles', which would include all four criteria together)—correspond to the simplest possible questions involving trade-offs that can ever be asked. They have the obvious advantage of being cognitively easier for people to choose from than full-profile choice sets and have been shown to reflect participants' true preferences more accurately [12, 13].

How a person responds to the choice tasks—i.e. their pairwise rankings of the hypothetical research proposals—reveals how they feel about the relative importance of the criteria with respect to the relative merits of alternative possible research proposals.

Such choice tasks, always involving a trade-off between the criteria, two at a time, are repeated with different pairs of hypothetical research proposals. Each time a participant ranks a pair of proposals, all other proposals that can be pairwise ranked by applying the logical property known as 'transitivity' are identified and eliminated, thereby minimising the number of choice tasks presented to each participant. For example, if a person prefers research proposal *A* to *B* and *B* to *C*, then, by transitivity, *A* is also preferred to *C*, and so this potential third choice task would not be presented to the person.

Thus, the DCE is adaptive in the sense that the sequence and set of choice tasks presented to each participant depends on their earlier answers. Because people's preferences are idiosyncratic and, to some extent, different from each other, people are likely to answer different numbers of questions each. Nonetheless, the PAPRIKA method ensures the number of choice sets is minimised for each person while guaranteeing they end up having pairwise ranked all

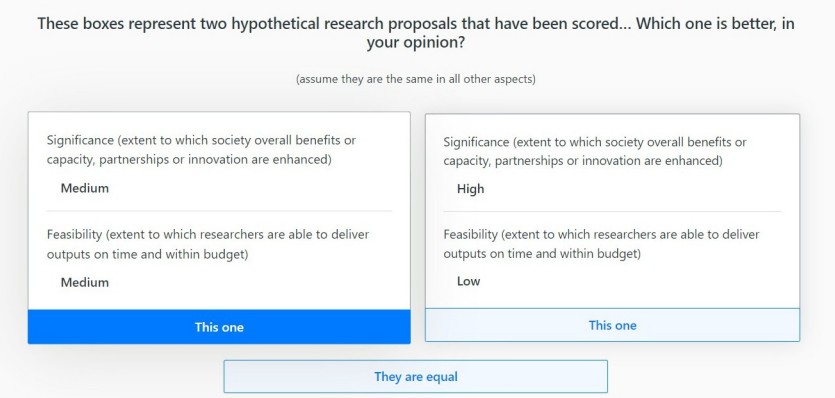

**Fig 1. An example of a choice task in 1000minds.**

possible research proposals defined on two criteria at a time, either explicitly or implicitly (by transitivity).

Two checks related to the quality of each participant's data are performed by the software. First, two choice tasks are repeated at the end of each participant's survey to check the consistency of their answers. Second, the software monitors how long each participant spent performing each choice task, as an indicator of how much thought they put into performing their DCE overall.

From each participant's pairwise rankings of the choice tasks included in the final dataset, the software uses quantitative methods based on linear programming to derive weights for the levels on each criteria, representing their relative importance; for technical details, see Hansen and Ombler [10]. As well as weights for each participant, the software reports the mean weights for groups of participants.

An additional criterion—monetary value of information (VOI) or return on investment (ROI)—was also considered as a potentially important determinant of research value. However, implementing ROI requires technical expertise to produce robust VOI estimates and there is a relative lack of familiarity with the concepts generally. There is also the risk of ROI being 'double-counted' with the *Significance* criterion. Therefore, instead of including it in the main research vehicle—a discrete choice experiment, as explained above—we decided to investigate stakeholders' opinions about ROI separately via specific survey questions.

The survey was pilot-tested before its online release by 5 researchers and 3 consumers, with minor refinements made to the final survey. Survey participants included members of ACTA (52 full member networks, 17 associate networks, 19 affiliate organisations and 194 individuals), ACTA Research Prioritisation Reference Group members, personal academic contacts of the Reference Group, research funders, consumers from ACTA project manager contacts and social media and Research Australia.

People on the email membership lists of the member networks and other sources listed above were emailed a link to the survey, which required them to express their consent as they started and to identify themselves as researchers/clinicians, consumers or funders. Ethics approval was provided by the University of Queensland Health and Behavioural Sciences, Low & Negligible Risk Ethics Sub-Committee (approval # 2020000828). The need for explicit consent was waived because the data collection was anonymous, non-sensitive and via the internet.

With respect to the analyses we performed, descriptive summaries consist of means, standard deviations and 95% confidence intervals for continuous data and frequency distributions for categorical data. Differences in participants' preferences, represented by the weights for the criteria, relative to their backgrounds (researcher, consumer, funder) were assessed using General Linear Models (GLM) for repeated measures with participant background as a between-subject factor. Estimated marginal means are reported to represent the average preference for each criterion, taking into account possible influences of participant background.

One-way ANOVA and Chi squared or Fisher's Exact tests, as appropriate, were used to make group comparisons of continuous and categorical data respectively. Significance levels were set at $\alpha = 0.05$. IBM SPSS version 27.0.1 (Armonk, NY) was used for data analysis.

## Results

There were 332 participants in the survey, of whom 137 (41%) were researchers, 176 (53%) were consumers and 19 (6%) were funders. Of the 137 researcher-participants, about half (66, 48%) were clinicians and the remainder were statisticians, operations managers or other research staff. Of the 176 consumer-participants, 96 (55%) had experience as a participant in a clinical trial and 92 (52%) had worked with researchers to design clinical research studies. Many of the researchers were highly experienced in clinical trials, with 82/137 (60%) having spent more than 10 years conducting or being involved in them. Most researcher-participants were involved in non-cancer research (74% non-cancer, 18% cancer, 6% cancer and other).

The 1000minds software automatically identifies participants eligible for exclusion from further analysis based on their unrealistically rapid responses: any choice task answered in less than 1 second or a median response time of less than 3 seconds. A similar majority of consumers (72%), researchers (80%) and funders (73%) answered the repeated choice tasks (reliability tests) consistently. Only 'non-speeder' participants who completed the full set of choice tasks were included in the final dataset of 220 participants for analysis: 104/176 (59%) consumers, 103/137 (75%) researchers and 15/19 (79%) funders (Table 1). Each person performed between 7 and 24 choice tasks, typically taking 5–15 minutes in total.

The mean (SD) raw weights for each criteria level are reported in Table 2. These weights can be used to score and rank research proposals by rating each proposal on the 4 criteria and summing the corresponding weights to get a total score on a scale from 0 to 100. For example, if a research proposal is evaluated by a committee that rates it as shown in Fig 2, its total score can used to rank it relative to other proposals.

GLM showed that the criteria are weighted differently overall (see Table 3, Wilk's lambda p<0.001). More weight is consistently given to the *Appropriateness* criterion (estimated marginal mean weight of 28.9% being greater by 2.8 to 9.4 points compared with the highest level of the other criteria, post hoc p = 0.035 to p<0.001). *Relevance* and *Significance* are weighted similarly (post hoc p = 0.51) and the least weight is goes to *Feasibility* (estimated marginal mean weight of 19.5 being less by 5.9 to 9.4 points compared with other criteria, post hoc p<0.001 for all comparisons).

Table 4 and Fig 3 shows the mean weights for each criterion-level of the research prioritisation framework, by participant background. There is a statistically significant association between criterion weight and respondent background (GLM, p<0.001) but the actual differences are very small. Consumers have similar weights to researchers and funders, except for the *Relevance* criterion which consumers weighted higher than researchers (mean difference of 2.7 points, post-hoc univariate ANOVA p = 0.037), and also *Feasibility* which consumers weight lower than researchers (mean difference 3.1 points, post hoc univariate ANOVA p = 0.037).

**Table 1. Response integrity.**

|  |  | Consumers | Researchers | Funders |
|---|---|---|---|---|
| Exclusions (N, %) * |  | 3 (1.7%) | 3 (2.2%) | 1 (5.3%) |
| Number of trade-off responses (median, range) | Completers (N) | 15 (7–24), n = 107 | 13 (7–22), n = 106 | 13 (7–15), n = 16 |
|  | Non-completers (N) | 1 (0–13), n = 69 | 2 (0–21), n = 31 | 0 (0–4), n = 3 |
| Completers (N) | Excluded | 3 | 3 | 1 |
|  | †Included | 104 | 103 | 15 |
| Consistency test among primary analysis cohort (n = 220) | 0 correct (n, %) | 3 (2.9%) | 0 | 0 |
|  | 1 correct (n, %) | 26 (25%) | 21 (20.4%) | 4 (26.7%) |
|  | 2 correct (n, %) | 75 (72.1%) | 82 (79.6%) | 11 (73.3%) |

*On the basis of responding to choice-tasks too quickly

†Primary analysis cohort

**Table 2. Mean weights (%) for the criteria and their levels.**

| Criteria | Mean (SD) | 95%CI |
|---|---|---|
| **Appropriateness** |  |  |
| *Low | 0 |  |
| Medium-low | 10.2 (4.0) | 9.7–10.8 |
| Medium | 18.7 (6.4) | 17.8–19.5 |
| Medium-high | 24.7 (7.5) | 23.7–25.7 |
| High | 29.3 (9.2) | 28.1–30.5 |
| **Significance** |  |  |
| Low | 0 |  |
| Medium-low | 9.1 (3.5) | 8.7–9.6 |
| Medium | 16.8 (5.5) | 16.0–17.5 |
| Medium-high | 22.3 (6.2) | 21.5–23.1 |
| High | 26.6 (7.2) | 25.6–27.5 |
| **Relevance** |  |  |
| Low | 0 |  |
| Medium-low | 8.8 (3.9) | 8.3–9.3 |
| Medium | 16.1 (6.2) | 15.2–16.9 |
| Medium-high | 21.2 (6.9) | 20.2–22.1 |
| High | 25.0 (7.5) | 24.0–26.0 |
| **Feasibility** |  |  |
| Low | 0 |  |
| Medium-low | 7.5 (3.9) | 7.0–8.0 |
| Medium | 13.3 (6.5) | 12.5–14.2 |
| Medium-high | 16.9(7.5) | 15.9–17.9 |
| High | 19.2 (8.2) | 18.1–20.2 |

Research proposals are scored by rating them on the 5 criteria and summing the 5 weights to get a total score (range 0–100).

* The lowest level of each criterion is set to zero by default.

**Fig 2. Example of ACTA research proposal evaluation.**

For researchers only (n = 103), there is no association between researcher characteristics (area of research, research role, experience of RCT involvement, experience of category 1 grant application) and their mean weights for any criteria (Table 5). For consumers only (n = 104), there are no significant differences in their mean weights for any criteria by experience of participating in a research study or working with researchers to design studies (Table 6). Funder-only analyses were not conducted because of the small number of funder participants.

Although there were many missing responses to the return on investment (ROI) questions, there does not appear to be a very strong view on whether ROI factors should be included in the research prioritisation process, with only about half of non-missing responses agreeing

**Table 3. Results of linear analysis (GLM).**

| | | Estimated marginal mean (95% CI) | Tests of within-subject effects | Tests of between-subject effects |
|---|---|---|---|---|
| Criteria | Appropriateness | 29 (27–31) | F = 19.2 (df 3), p<0.001 | NA |
| | Significance | 26 (25–28) | | |
| | Relevance | 25 (24–27) | | |
| | Feasibility | 20 (18–21) | | |
| Interaction term | Criterion by Respondent Category | Not displayed | F = 2.56 (df 6), p = 0.02 | NA |

*Wilks' Lambda <0.001; df = degrees of freedom.

**Table 4. Mean criteria weights (%) by respondent group.**

| Group: | Consumers *n = 104* | | Researchers *n = 103* | | Funders *n = 15* | | |
|---|---|---|---|---|---|---|---|
| Criterion: | Mean (SD) | 95%CI | Mean (SD) | 95%CI | Mean (SD) | 95%CI | p-value* |
| **Appropriateness** | | | | | | | |
| †Low | 0 | | 0 | | 0 | | |
| Low-medium | 9.5 (4.1) | 8.7–10.3 | 10.9 (3.8) | 10.2–11.6 | 10.5 (3.6) | 8.5–12.5 | |
| Medium | 17.6 (6.7) | 16.3–18.9 | 19.8 (6.1) | 18.6–20.9 | 18.8 (5.4) | 15.8–21.7 | |
| High-medium | 23.7 (7.7) | 22.2–25.2 | 25.8 (7.3) | 24.3–27.2 | 24.2 (6.2) | 20.7–27.6 | |
| High | 28.6 (9.4) | 26.8–30.4 | 30.2 (9.1) | 28.4–31.9 | 28.1 (8.2) | 23.5–32.6 | 0.43 |
| **Significance** | | | | | | | |
| Low | 0 | | 0 | | 0 | | |
| Low-medium | 9.4 (3.7) | 8.7–10.1 | 9.5 (3.3) | 8.4–9.7 | 7.8 (2.7) | 6.3–9.3 | |
| Medium | 17.3 (5.8) | 16.1–18.4 | 16.5 (5.3) | 15.5–17.6 | 14.7 (4.5) | 12.3–17.2 | |
| High-medium | 23.0 (6.5) | 21.8–24.3 | 21.8 (6.0) | 20.6–23.0 | 20.3 (4.9) | 17.6–23.0 | |
| High | 27.6 (7.5) | 26.1–29.1 | 25.8 (6.9) | 24.4–27.1 | 24.9 (5.1) | 22.1–27.8 | 0.12 |
| **Relevance** | | | | | | | |
| Low | 0 | | 0 | | 0 | | |
| Low-medium | 9.4 (4.5) | 8.6–10.3 | 8.2 (3.2) | 7.5–8.8 | 9.0 (2.8) | 7.5–10.5 | |
| Medium | 17.1 (7.2) | 15.7–18.5 | 14.9 (5.2) | 13.9–15.9 | 16.6 (4.4) | 14.1–19.0 | |
| High-medium | 22.4 (7.9) | 20.8–23.9 | 19.8 (5.7) | 18.7–20.9 | 22.1 (5.3) | 19.2–25.1 | |
| High | 26.2 (8.4) | 24.6–27.9 | 23.5 (6.3) | 22.3–24.7 | 26.5 (6.9) | 22.7–30.3 | ‡0.02 |
| **Feasibility** | | | | | | | |
| Low | 0 (0) | | | | | | |
| Low-medium | 6.6 (3.8) | 5.8–7.3 | 8.4 (3.9) | 7.7–9.2 | 7.9 (3.7) | 5.9–10.0 | |
| Medium | 11.8 (6.4) | 10.5–13.0 | 14.8 (6.4) | 13.5–16.0 | 14.1 (5.8) | 10.9–17.3 | |
| High-medium | 15.2 (7.6) | 13.7–26.7 | 18.4 (7.4) | 17.0–19.9 | 17.9 (6.2) | 14.5–21.3 | |
| High | 17.5 (8.5) | 15.9–19.2 | 20.6 (7.9) | 19.1–22.1 | 20.5 (6.3) | 17.0–24.0 | ‡0.02 |

*Overall p-value for one-way ANOVA, comparing the mean weight for the highest category by respondent group, in each criterion separately.

†By definition, the lowest level of each criterion is set to zero.

‡Post-hoc test showed a significant difference between researchers and consumers (p = 0.037).

with that proposition (Table 7). However, there is a clear view that if ROI is to be included in a prioritisation process, then it should be in combination with other factors and not as a stand-alone consideration (nearly 90% of non-missing responses). A guideline and web-based tool appears to be the most endorsed means of assisting with technical calculation of ROI.

## Discussion

Although scoring systems involving multiple criteria are commonly used to evaluate research proposals, such as in grant funding decisions, such systems have not previously been weighted using formal stated-preference techniques. Using a discrete choice experiment, we found that four previously identified criteria thought to be associated with the relative merit of research proposals were weighted fairly similarly by consumers, researchers and funders. Although the differences in weights across criteria was small, the most important criterion was *Appropriateness* (estimated marginal mean, taking into account differences between respondent background, was 29) and the least important was *Feasibility* (estimated marginal mean 20).

The main implication of this study is that the proposed scoring framework provides an objective and pragmatic tool for ranking research proposals, incorporating the views of a

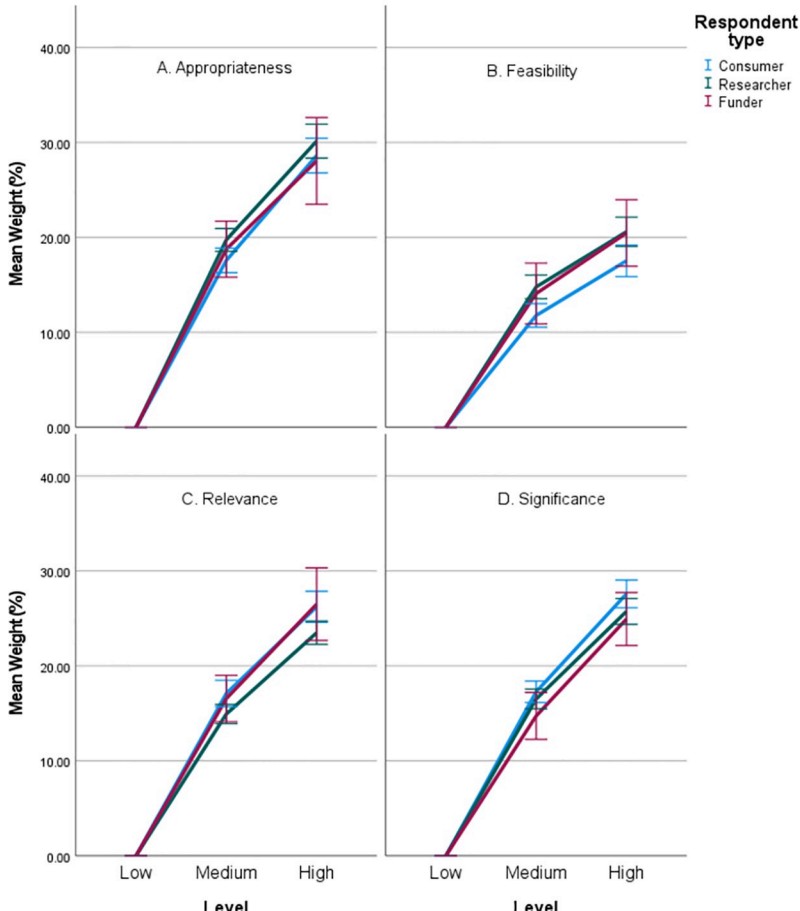

**Fig 3. Mean criteria weights (%) by participant category.** Panel A Appropriateness; Panel B Feasibility; Panel C Relevance; Panel D Significance; error bars 95% CI.

range of stakeholders including researchers, consumers and funders. Furthermore, the minimal differences in criteria weights across stakeholder groups provides reassurance that the preferences of these groups are fairly represented in the tool. The relatively higher weight for *Appropriateness* emphasises the importance for researchers and funders to look carefully at the research question and ensure that the chosen study design is adequate to effectively produce an answer. The relatively lower weight for *Feasibility* suggests that whether a research study *can* be done is not as important as whether it *should* be done. Finally, the total score calculated by summing the weights across the criteria can be compared with (divided by) the cost of research to arrive at, in effect, a merit-cost ratio.

There are some limitations to the study. First, the response rate of potential participants could not be estimated due to the broad-based, snow-ball approach to recruiting participants, and the lack of a known denominator for some stakeholder groups. Also, there was a substantial proportion of incomplete responses, especially from consumers. It is not obvious how that would bias the observation and it is possible that participants who completed the survey gave a more thoughtful and meaningful response. The number of funder-participants may be insufficient to draw strong conclusions on the distribution of their weightings. In addition, the extent to which all relevant stakeholders, in particular minority populations or those with inequitable

**Table 5. Criteria weights (as %) by researcher characteristics (N = 103).**

| Characteristic (n) | | Criteria weight (highest category) | | | |
|---|---|---|---|---|---|
| | | **Appropriateness** | **Significance** | **Relevance** | **Feasibility** |
| Area of work | Cancer (20) | 33.5 (8.3) | 25.9 (5.4) | 21.0 (8.9) | 19.7 (9.2) |
| | Other (79) | 29.4 (7.8) | 25.9 (7.4) | 24.0 (5.4) | 20.7 (7.8) |
| | Cancer + other (4) | 29.4 (6.7) | 24.8 (4.0) | 25.9 (5.4) | 22.9 (2.3) |
| * p-value | | 0.20 | 0.52 | 0.12 | 0.73 |
| Experience of RCT involvement (years) | Less than 5 years (31) | 32.2 (10.4) | 25.1 (6.1) | 22.2 (7.5) | 20.4 (8.3) |
| | 5 to 10 years (10) | 33.9 (10.1) | 21.8 (4.7) | 25.0 (4.1) | 19.4 (9.1) |
| | More than 10 years (62) | 28.5 (7.9) | 26.7 (7.4) | 23.9 (5.9) | 20.9 (7.6) |
| p-value | | 0.07 | 0.10 | 0.35 | 0.84 |
| Experience of category 1 grant applications † | None (25) | 30.7 (9.2) | 24.8 (6.4) | 23.7 (7.5) | 20.8 (7.9) |
| | Less than 2 years (16) | 30.0 (10.9) | 27.9 (6.2) | 23.0 (7.1) | 19.1 (9.1) |
| | 2 to 5 years (14) | 28.7 (5.0) | 26.9 (5.3) | 23.3 (5.3) | 20.9 (6.4) |
| | Greater than 5 years (48) | 30.3 (9.5) | 25.2 (7.8) | 23.6 (5.7) | 20.9 (8.0) |
| p-value | | 0.93 | 0.44 | 0.98 | 0.88 |

*Overall p-value for one-way ANOVA, comparing the mean weight for the highest category by respondent group, in each criterion separately.

†Category 1 grants are from major government-funders such as the National Health and Medical Research Council or Medical Research Futures Fund.

access to healthcare and health decision-making, were represented by the participants in this study may be insufficient. It is possible that the preferences of such groups are different to what we observed.

Further work is required to confirm the reliability of rating research proposals on the criteria during a proposal evaluation process, and to validate the resulting ranking of total scores

**Table 6. Criteria weights (%) by consumer characteristics (N = 104).**

| Characteristic (n) | | Criteria weight (highest category, mean SD) | | | |
|---|---|---|---|---|---|
| | | **Appropriateness** | **Significance** | **Relevance** | **Feasibility** |
| Area of work | Cancer (50) | 26.5 (5.4) | 26.1 (5.5) | 27.2 (5.3) | 20.4 (7.3) |
| | Non-cancer (31) | 26.5 (6.6) | 27.5 (6.5) | 25.6 (8.4) | 20.4 (8.3) |
| | Cancer + other (2) | 25.4 (0.5) | 25.4 (0.5) | 28.2 (4.6) | 21.1 (5.6) |
| | Missing (93) | 27.8 (9.2) | 26.8 (6.7) | 25.4 (7.4) | 20.1 (8.2) |
| * p-value | | 0.71 | 0.77 | 0.50 | 0.99 |
| Experience of participating in research studies | None (78) | 26.4 (6.2) | 27.1 (6.5) | 25.5 (6.1) | 21.1 (7.2) |
| | Less than 1 year (34) | 28.2 (9.2) | 24.5 (5.4) | 27.5 (9.2) | 19.8 (8.6) |
| | 1–5 years (46) | 27.9 (9.8) | 27.4 (6.9) | 25.7 (7.2) | 19.1 (8.5) |
| | More than 5 years (16) | 27.4 (5.2) | 27.5 (4.6) | 25.4 (6.0) | 19.8 (8.2) |
| | Missing (2) | 23.0 (2.8) | 2.7 (0.9) | 29.6 (6.5) | 21.7 (4.7) |
| p-value | | 0.66 | 0.25 | 0.62 | 0.73 |
| Experience of working with researchers to design studies | Less than 1 year (20) | 28.8 (10.7) | 25.6 (3.5) | 27.6 (8.1) | 18.0 (7.6) |
| | 1–5 years (43) | 25.7 (7.3) | 28.7 (6.1) | 24.9 (6.8) | 20.8 (9.1) |
| | More than 5 years (29) | 27.7 (6.2) | 25.0 (6.0) | 27.7 (6.8) | 19.6 (6.8) |
| | None (84) | 27.4 (7.7) | 26.5 (6.8) | 25.5 (7.0) | 20.7 (7.6) |
| p-value | | 0.45 | 0.07 | 0.23 | 0.51 |

*Overall p-value from separate (for each criteria) one-way ANOVA with criteria weight as the dependent variable.

**Table 7. Responses to return on investment (ROI) questions (n, %).**

| Question | | Response | Consumers (N = 176) | Researchers (N = 137) | Funders (N = 19) |
|---|---|---|---|---|---|
| Should ROI be included in the prioritisation process? | | Yes | 59 (33.5) | 60 (43.8) | 8 (42.1) |
| | | No | 20 (11.4) | 26 (19.0) | 4 (21.1) |
| | | Unsure | 27 (15.3) | 20 (14.6) | 3 (15.8) |
| | | Missing | 70 (39.8) | 31 (22.6) | 2 (10.5) |
| How should ROI be incorporated into prioritisation process? | | The only factor | 5 (2.8) | 6 (4.4) | 2 (10.5) |
| | | In combination with merit framework | 95 (54.0) | 93 (67.9) | 13 (68.4) |
| | | Unsure | 6 (3.4) | 7 (5.1) | 0 |
| | | Missing | 70 (39.8) | 31 (22.6) | 4 (21.1) |
| Ways that might be helpful in calculating return on investment | Training workshop | Yes | 61 (34.7) | 53 (38.7) | 8 (42.1) |
| | | No | 45 (25.6) | 49 (35.8) | 7 (36.8) |
| | | Missing | 70 (39.8) | 35 (25.5) | 4 (21.1) |
| | Web-based tool | Yes | 61 (34.7) | 66(48.2) | 9 (47.4) |
| | | No | 45 (25.6) | 36 (26.3) | 6 (31.6) |
| | | Missing | 70 (39.8) | 35(25.5) | 4 (21.1) |
| | Guideline | Yes | 87 (49.4) | 82 (59.9) | 14 (73.7) |
| | | No | 19 (10.8) | 20 (14.6) | 1 (5.3) |
| | | Missing | 70 (39.8) | 35 (25.5) | 4 (21.1) |
| | Other | Yes | 9 (5.1) | 5 (3.6) | 1 (5.3) |
| | | No | 97 (55.1) | 97 (70.8) | 14 (73.7) |
| | | Missing | 70 (39.8) | 35 (25.5) | 4 (21.1) |

ROI = return on investment.

against markers of important research. Also, it is unclear from this research whether stakeholders see return on investment as a necessary aspect for determining the worthiness of a research proposal; however, this requires further investigation due to the relatively low participation in the study by research funders and high number of missing responses.

## Author Contributions

**Conceptualization:** Haitham Tuffaha, Carmel M. Hawley, Philip Peyton, Alisa M. Higgins, Paul A. Scuffham, Rachael L. Morton.

**Formal analysis:** William J. Taylor, Paul Hansen, Angela Jacques, Rachael L. Morton.

**Investigation:** Haitham Tuffaha, Philip Peyton, Alisa M. Higgins, Paul A. Scuffham, Fiona Nemeh, Anitha Balagurunathan, Rachael L. Morton.

**Methodology:** William J. Taylor, Haitham Tuffaha, Paul Hansen, Angela Jacques.

**Project administration:** Fiona Nemeh, Anitha Balagurunathan, Rachael L. Morton.

**Software:** Paul Hansen.

**Supervision:** Carmel M. Hawley.

**Writing – original draft:** William J. Taylor.

**Writing – review & editing:** Haitham Tuffaha, Carmel M. Hawley, Philip Peyton, Alisa M. Higgins, Paul A. Scuffham, Fiona Nemeh, Paul Hansen, Angela Jacques, Rachael L. Morton.

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
