## [Decision Letter · Decision Letter 0]

12 Dec 2022

PONE-D-22-25100Embedding stakeholder preferences in setting priorities for health researchPLOS ONE

Dear Dr. Taylor,

Thank you for submitting your manuscript to PLOS ONE. After careful consideration, we feel that it has merit but does not fully meet PLOS ONE’s publication criteria as it currently stands. Therefore, we invite you to submit a revised version of the manuscript that addresses the points raised during the review process.

We look forward to receiving your revised manuscript.

Kind regards,

Gian Mauro Manzoni, Ph.D., Psy.D.

Academic Editor

PLOS ONE

Journal Requirements:

2.  Please provide additional details regarding participant consent. In the ethics statement in the Methods and online submission information, please ensure that you have specified what type you obtained (for instance, written or verbal, and if verbal, how it was documented and witnessed). If your study included minors, state whether you obtained consent from parents or guardians. If the need for consent was waived by the ethics committee, please include this information

Additional Editor Comments (if provided):

Reviewer 1: Thank you for the opportunity to review this paper applying a discrete choice experiment to derive empirical weights for criteria evaluating research proposals. The work will be of interest to anyone involved in prioritising research spending, where evidence for personal or organisational weights is limited.

I have organised my comments by PLOS One’s criteria.

1. Technical soundness

The data as presented support the conclusions and the limitations are highlighted; however, I think it would be helpful to provide additional information in the Methods to help readers evaluate the limitations. Specifically: i) could you comment on the suitability of the sample size, or your target sample size, for example using a typical rule-of-thumb estimate? ii) to address response rate, could you perhaps give a sense of how many initial invitation emails were sent out, or other measure of recruitment scale?

2. Statistical methods

Please could you expand a little on the statistical methods, for readers not familiar with 1000minds and PAPRIKA. As a reader I would value a summary rather than having to read the referenced paper. It wasn’t clear to me the rationale for trading only two attributes in each choice set, how the adaptive choice sets work, nor how and why the completers ended up being presented with different numbers of choice questions (or am I misunderstanding the numbers?). I would also like to see a statement of the statistical analysis used to derive the weights – for example is it basically a conditional logit, were random effects included? As written the analysis is something of a black box, which makes it hard for the reader to evaluate.

3. Clarity

Overall the paper is clearly written; I have some minor suggestions and queries that might help the reader.

Intro:

Paragraph beginning ‘Using this framework’ – it would help to state clearly here that this paper reports the derivation of the weights, using a discrete choice experiment. That would make it clear what’s been done before, vs what this paper is – and would also make sense of the introduction of the DCE in the final paragraph, which otherwise is a bit of a surprise.

Methods:

Check references – for example, text gives Hansen and Ombler as 9, but it’s reference 10 in the list.

Can you be absolutely clear who was excluded (the speeders?) and who were retained (the inconsistent?). Either here or in the Results.

Results:

Table 1: Completers, Funders, excluded – should that be 1 not 2?

It’s not obvious to me how the p-value in the footnote to Table 4 relates to the p-values for Relevance and Feasibility given in the text

Personally, I would prefer to see the total N for Table 5, Table 6 and the columns in Table 7 – I know it’s in other tables, or I could add them up, but it just makes life easier for the reader.

There is no funder-only analysis, because of small sample size? Would be helpful to state.

Reviewe 2: Thank you for the opportunity to read the paper “embedding stakeholder preferences in setting priorities for health research”. The following are comments to support the paper further.

The paper is well written and was a real pleasure to read. The authors should be commended for the research, and this will likely be of interest to many readers of this journal.

Title: One main recommendation is to change the title of this paper as it does not truly reflect the content. Perhaps consider more details a “embedding stakeholder preferences in setting priorities for health research: using a discrete choice experiment to develop a multi-criteria tool for evaluating research proposals.” Or “Towards the development of a multi-criterial tool for evaluating research proposals that reflects stakeholder preferences.”. The title is a tad misleading as it stands now.

Abstract: No changes suggested

Introduction:

- Paragraph 1, please define “low value published studies”. Please consider non-colonial approaches to how value can be interpreted.

- The last paragraph of the introduction may belong more in the discussion. Your last paragraph is a strong lead into the methods section, and this paragraph is someone distracting to the reader.

Methods:

- Third paragraph, please define consumer.

- Remainder of methods was easy to follow and reproducible. The level of detail was refreshing for a DCE paper.

- Only area related to the statistical analysis. Did the authors make any adjustments for multiple comparisons?

Results:

Very well written and presented.

Discussion:

Once again very well written. In the last paragraph, the authors are encouraged to consider future research that considers equity and diversity. In most research, many voices are missing at the table. This paper is an excellent beginning to have a tool to consider how embedding stakeholder preferences can happen. But the authors are encouraged to consider how weights may change based on who the stakeholders/collaborators are. Weighting of certain factors may be different, and this is an important area of future research and awareness for our academic community.

Figure 1: figure is blurry and could benefit from some graphical support to ensure it is crisp. Recognize that this may only be a problem with how we are accessing the document as peer reviewers.

Reviewer3: While it is interesting and valuable to undertake participatory decision processes within the clinical context, I am sceptical as to how the application of the PAPRIKA method has been done in this manuscript. In particular, the calibration of the purely ordinal performance levels into "Low", "Medium", "High" in relation to the rather abstract set of criteria that is not quite operationalised before put into concrete questions to be answered by respondents. The meaning of "Low" etc. is not really made clear and it cannot be understood beforehand to be universal between respondents, however this could be remediated by providing baselines values or anchoring values that puts the respodents on the "same page" when they answer the choice experiment otherwise the resulting weights will suffer from lack of meaning.

Perhaps what could be done, but then with a fewer set of respondents, is to connect the performance labels "Low, Medium, High" to previous projects and making them more concrete for the respondents to react upon. You'd want the respondents to be aware of what the trade-off actually is about, what you get and don't get from an individual sequence of trade-offs.

I am not sure about the data availability part. I know the the 1000minds platform will enable for you to access your data rather swiftly but it is unclear if the data is made accessible to other than the platform license holder in this case.

Reviewers' comments:

Reviewer's Responses to Questions

**Comments to the Author**

1. Is the manuscript technically sound, and do the data support the conclusions?

Reviewer #1: Partly

Reviewer #2: Yes

Reviewer #3: Partly

2. Has the statistical analysis been performed appropriately and rigorously? 

Reviewer #1: I Don't Know

Reviewer #2: Yes

Reviewer #3: No

3. Have the authors made all data underlying the findings in their manuscript fully available?

Reviewer #1: Yes

Reviewer #2: Yes

Reviewer #3: No

4. Is the manuscript presented in an intelligible fashion and written in standard English?

Reviewer #1: Yes

Reviewer #2: Yes

Reviewer #3: Yes

5. Review Comments to the Author

Reviewer #1: Thank you for the opportunity to review this paper applying a discrete choice experiment to derive empirical weights for criteria evaluating research proposals. The work will be of interest to anyone involved in prioritising research spending, where evidence for personal or organisational weights is limited.

I have organised my comments by PLOS One’s criteria.

1. Technical soundness

The data as presented support the conclusions and the limitations are highlighted; however, I think it would be helpful to provide additional information in the Methods to help readers evaluate the limitations. Specifically: i) could you comment on the suitability of the sample size, or your target sample size, for example using a typical rule-of-thumb estimate? ii) to address response rate, could you perhaps give a sense of how many initial invitation emails were sent out, or other measure of recruitment scale?

2. Statistical methods

Please could you expand a little on the statistical methods, for readers not familiar with 1000minds and PAPRIKA. As a reader I would value a summary rather than having to read the referenced paper. It wasn’t clear to me the rationale for trading only two attributes in each choice set, how the adaptive choice sets work, nor how and why the completers ended up being presented with different numbers of choice questions (or am I misunderstanding the numbers?). I would also like to see a statement of the statistical analysis used to derive the weights – for example is it basically a conditional logit, were random effects included? As written the analysis is something of a black box, which makes it hard for the reader to evaluate.

4. Clarity

Overall the paper is clearly written; I have some minor suggestions and queries that might help the reader.

Intro:

Paragraph beginning ‘Using this framework’ – it would help to state clearly here that this paper reports the derivation of the weights, using a discrete choice experiment. That would make it clear what’s been done before, vs what this paper is – and would also make sense of the introduction of the DCE in the final paragraph, which otherwise is a bit of a surprise.

Methods:

Check references – for example, text gives Hansen and Ombler as 9, but it’s reference 10 in the list.

Can you be absolutely clear who was excluded (the speeders?) and who were retained (the inconsistent?). Either here or in the Results.

Results:

Table 1: Completers, Funders, excluded – should that be 1 not 2?

It’s not obvious to me how the p-value in the footnote to Table 4 relates to the p-values for Relevance and Feasibility given in the text

Personally, I would prefer to see the total N for Table 5, Table 6 and the columns in Table 7 – I know it’s in other tables, or I could add them up, but it just makes life easier for the reader.

There is no funder-only analysis, because of small sample size? Would be helpful to state.

Reviewer #2: Thank you for the opportunity to read the paper “embedding stakeholder preferences in setting priorities for health research”. The following are comments to support the paper further.

The paper is well written and was a real pleasure to read. The authors should be commended for the research, and this will likely be of interest to many readers of this journal.

Title: One main recommendation is to change the title of this paper as it does not truly reflect the content. Perhaps consider more details a “embedding stakeholder preferences in setting priorities for health research: using a discrete choice experiment to develop a multi-criteria tool for evaluating research proposals.” Or “Towards the development of a multi-criterial tool for evaluating research proposals that reflects stakeholder preferences.”. The title is a tad misleading as it stands now.

Abstract: No changes suggested

Introduction:

- Paragraph 1, please define “low value published studies”. Please consider non-colonial approaches to how value can be interpreted.

- The last paragraph of the introduction may belong more in the discussion. Your last paragraph is a strong lead into the methods section, and this paragraph is someone distracting to the reader.

Methods:

- Third paragraph, please define consumer.

- Remainder of methods was easy to follow and reproducible. The level of detail was refreshing for a DCE paper.

- Only area related to the statistical analysis. Did the authors make any adjustments for multiple comparisons?

Results:

Very well written and presented.

Discussion:

Once again very well written. In the last paragraph, the authors are encouraged to consider future research that considers equity and diversity. In most research, many voices are missing at the table. This paper is an excellent beginning to have a tool to consider how embedding stakeholder preferences can happen. But the authors are encouraged to consider how weights may change based on who the stakeholders/collaborators are. Weighting of certain factors may be different, and this is an important area of future research and awareness for our academic community.

Figure 1: figure is blurry and could benefit from some graphical support to ensure it is crisp. Recognize that this may only be a problem with how we are accessing the document as peer reviewers.

Reviewer #3: While it is interesting and valuable to undertake participatory decision processes within the clinical context I am sceptical as to how the application of the PAPRIKA method has been done in this manuscript. In particular, the calibration of the purely ordinal performance levels into "Low", "Medium", "High" in relation to the rather abstract set of criteria that is not quite operationalised before put into concrete questions to be answered by respondents. The meaning of "Low" etc. is not really made clear and it cannot be understood beforehand to be universal between respondents, however this could be remediated by providing baselines values or anchoring values that puts the respodents on the "same page" when they answer the choice experiment otherwise the resulting weights will suffer from lack of meaning.

Perhaps what could be done, but then with a fewer set of respondents, is to connect the performance labels "Low, Medium, High" to previous projects and making them more concrete for the respondents to react upon. You'd want the respondents to be aware of what the trade-off actually is about, what you get and don't get from an individual sequence of trade-offs.

I am not sure about the data availability part. I know the the 1000minds platform will enable for you to access your data rather swiftly but it is unclear if the data is made accessible to other than the platform license holder in this case.

6. PLOS authors have the option to publish the peer review history of their article (what does this mean?). If published, this will include your full peer review and any attached files.

Reviewer #1: No

Reviewer #2: No

Reviewer #3: No

---

## [Author Response · Author response to Decision Letter 0]

13 Apr 2023

Editorial comment

Response: The revised manuscript has been formatted to meet style requirements. Change to manuscript: Minor editing changes throughout.

2. Please provide additional details regarding participant consent. 

Response: The need for explicit consent was waived by the ethics committee since the data collection was non-sensitive, anonymous and via internet. This information is recorded in the ethics statement. Change to manuscript: P11 line 16-17: “The need for explicit consent was waived since the data collection was anonymous, non-sensitive and via internet.”

Reviewer 1

1. I think it would be helpful to provide additional information in the Methods to help readers evaluate the limitations. Specifically: i) could you comment on the suitability of the sample size, or your target sample size, for example using a typical rule-of-thumb estimate? ii) to address response rate, could you perhaps give a sense of how many initial invitation emails were sent out, or other measure of recruitment scale 

Response: The desirable sample size was not precalculated for comparing the utility weights between subgroups; a post hoc power calculation for ANOVA with 4 groups, power 80% and alpha 5% shows a small to medium effect (ES 0.225) can be detected with 220 participants. Since it is doubtful that such a calculation is useful, we have not chosen to add this information to the manuscript. We are not able to estimate response rate since the initial email and web-based invitations were distributed to an unknown denominator. Changes to the manuscript: No changes made.

2. Please could you expand a little on the statistical methods, for readers not familiar with 1000minds and PAPRIKA. As a reader I would value a summary rather than having to read the referenced paper. It wasn’t clear to me the rationale for trading only two attributes in each choice set, how the adaptive choice sets work, nor how and why the completers ended up being presented with different numbers of choice questions (or am I misunderstanding the numbers?). I would also like to see a statement of the statistical analysis used to derive the weights – for example is it basically a conditional logit, were random effects included? As written the analysis is something of a black box, which makes it hard for the reader to evaluate. 

Response: Additional information about the PAPRIKA method is now provided (and please see the next column). (1) The rationale for just two criteria in each choice task is now explained, as is (2) the adaptive algorithm, and (3) why participants are presented with different numbers of choice tasks. Finally, (4) the quantitative methods used for determining the weights are now expanded on. We have done our best to lift the lid on the black box, while staying with word limits. Thank you for your questions. Changes to the manuscript: (1), page 9 line 4-9: “These choice tasks, based on just two criteria at a time – known as ‘partial profiles’ (in contrast to ‘full profiles’, which would include all four criteria together) – correspond to the simplest possible questions involving trade-offs that can ever be asked. They have the obvious advantage of being cognitively easier for people to choose from than full-profile choice sets, and have been shown to reflect participants’ true preferences more accurately (Chrzan 2010; Meyerhoff & Oehlmann 2023).” (2 & 3), page 9 line 14-21, page 10 line 2-7: “Such choice tasks, always involving a trade-off between the criteria, two at a time, are repeated with different pairs of hypothetical research proposals. Each time a participant ranks a pair of proposals, all other proposals that can be pairwise ranked by applying the logical property known as ‘transitivity’ are identified and eliminated, thereby minimising the number of choice tasks presented to each participant. For example, if a person prefers research proposal A to B and B to C, then, by transitivity, A is also preferred to C, and so this potential third choice task would not be presented to the person. Thus, the DCE is adaptive in the sense that the sequence and set of choice tasks presented to each participant depends on their earlier answers. Because people’s preferences are idiosyncratic and, to some extent, different from each other, people are likely to answer different numbers of questions each. Nonetheless, the PAPRIKA method ensures the number of choice sets is minimised for each person while guaranteeing they end up having pairwise ranked all possible research proposals defined on two criteria at a time, either explicitly or implicitly (by transitivity).” (4), page 10 line 13-16: “From each participant’s pairwise rankings of the choice tasks included in the final dataset, the software uses quantitative methods based on linear programming to derive weights for the levels on each criteria, representing their relative importance; for technical details, see Hansen and Ombler (10).”

3. Paragraph beginning ‘Using this framework’ – it would help to state clearly here that this paper reports the derivation of the weights, using a discrete choice experiment. That would make it clear what’s been done before, vs what this paper is – and would also make sense of the introduction of the DCE in the final paragraph, which otherwise is a bit of a surprise.

Response: We have revised this paragraph to emphasize that the study is mainly concerned with derivation of weights using a DCE as suggested. Changes to manuscript: Page 6 line 10 additional text: “(specifically, a discrete choice experiment)”

4. Check references – for example, text gives Hansen and Ombler as 9, but it’s reference 10 in the list. 

Response: Thank you for identifying these errors, which are now corrected. Changes to manuscript: Page 10 line 16: In text citation change to (10)

5. Can you be absolutely clear who was excluded (the speeders?) and who were retained (the inconsistent?). Either here or in the Results.

Response: Only ‘speeders’ and non-completers were excluded. Changes to the manuscript: Page 13 line 1-3, adjusted text: “Only ‘non-speeder’ participants who completed the full set of choice-tasks were included in the final dataset of 220 participants for analysis…”

6. Table 1: Completers, Funders, excluded – should that be 1 not 2?

Response: Yes – thank you for identifying that error. Changes to the manuscript: Table 1 corrected.

7. It’s not obvious to me how the p-value in the footnote to Table 4 relates to the p-values for Relevance and Feasibility given in the text

Response: Yes – thank you for identifying that error. The overall p-values for the ANOVA, rather than the p-value from the post-hoc test are shown in the text. Changes to manuscript: Page 14 line 3 and 5 correct p-value now shown

8. Personally, I would prefer to see the total N for Table 5, Table 6 and the columns in Table 7 – I know it’s in other tables, or I could add them up, but it just makes life easier for the reader.

Response: Total N for these Tables added as requested Changes to manuscript: Total N added to Tables 5-7

9. There is no funder-only analysis, because of small sample size? Would be helpful to state. 

Response: This is correct. Changes to manuscript: Page 14 line 15-16: “Funder-only analyses were not conducted because of the small number of funder participants.”

Reviewer 2

1. One main recommendation is to change the title of this paper as it does not truly reflect the content. Perhaps consider more details a “embedding stakeholder preferences in setting priorities for health research: using a discrete choice experiment to develop a multi-criteria tool for evaluating research proposals.” Or “Towards the development of a multi-criterial tool for evaluating research proposals that reflects stakeholder preferences.”. The title is a tad misleading as it stands now. 

Response: We have changed the title as suggested. Changes to manuscript: Title changed to “Embedding stakeholder preferences in setting priorities for health research: using a discrete choice experiment to develop a multi-criteria tool for evaluating research proposals”

2. Paragraph 1, please define “low value published studies”. Please consider non-colonial approaches to how value can be interpreted. 

Response: “Value” can clearly be considered in different ways, but one overall concept in the field of treatment for health conditions, is that the study should lead to a change in health care practice that creates benefit for the people receiving that health care. We are uncertain how the reviewer wishes the notion of ‘colonialism’ to be incorporated here. Changes made to the manuscript: No changes made to the manuscript.

3. The last paragraph of the introduction may belong more in the discussion. Your last paragraph is a strong lead into the methods section, and this paragraph is someone distracting to the reader.

Response: We have suggested moving the paragraph into the methods section rather than the discussion section, and hope this is acceptable to the reviewer. Changes made to the manuscript: Page 10 lines 18-22; page 11 lines 1-3: Paragraph from introduction section moved to methods section

4. Third paragraph, please define consumer. 

Response: ‘Consumer’ is defined as a user of health care services. Changes made to the manuscript: Page 8 line 2: addition text “(defined as users of healthcare services)”

5. Only area related to the statistical analysis. Did the authors make any adjustments for multiple comparisons?

Response: No adjustments were made for multiple comparisons. Changes made to the manuscript: No changes were made.

6. In the last paragraph, the authors are encouraged to consider future research that considers equity and diversity. In most research, many voices are missing at the table. This paper is an excellent beginning to have a tool to consider how embedding stakeholder preferences can happen. But the authors are encouraged to consider how weights may change based on who the stakeholders/collaborators are. Weighting of certain factors may be different, and this is an important area of future research and awareness for our academic community. 

Response: Additional text added to the discussion as suggested. Changes made to the manuscript: Page 16 lines 11-15: additional text “In addition, the extent to which all relevant stakeholders, in particular minority populations or those with inequitable access to healthcare and health decision-making, were represented by the participants in this study may be insufficient. It is possible that the preferences of such groups are different to what we observed.”

Fig 1 is blurry 

Response: We have hopefully created a clearer version. Changes made to the manuscript: Figure 1 updated

Reviewer 3

1. I am sceptical as to how the application of the PAPRIKA method has been done in this manuscript. In particular, the calibration of the purely ordinal performance levels into "Low", "Medium", "High" in relation to the rather abstract set of criteria that is not quite operationalised before put into concrete questions to be answered by respondents. The meaning of "Low" etc. is not really made clear and it cannot be understood beforehand to be universal between respondents, however this could be remediated by providing baselines values or anchoring values that puts the respondents on the "same page" when they answer the choice experiment otherwise the resulting weights will suffer from lack of meaning. Perhaps what could be done, but then with a fewer set of respondents, is to connect the performance labels "Low, Medium, High" to previous projects and making them more concrete for the respondents to react upon. You'd want the respondents to be aware of what the trade-off actually is about, what you get and don't get from an individual sequence of trade-offs. 

Response: It is not feasible within the limitations of this particular manuscript to repeat the data collection with a revised framework suggested by the reviewer. Moreover, as we now explain (please see the next column), it is unnecessary to have a concrete anchor within each dimension, for the relative weights to be meaningful within the responses from an individual. Changes made to the manuscript: We now say (page 8 lines 10-16): “It is worthwhile noting that how participants interpret these levels in the DCE is relative to their own personal experiences and not intended to be interpersonally comparable. For example, what one person thinks is a medium performance on any of the four criteria could be a low performance for another person or a high performance for another, but that is irrelevant here because the objective is to discover the criteria’s relative importance to each individual. As mentioned above, an important strength of the PAPRIKA method is that a set of weights is generated for each individual.”

2. I am not sure about the data availability part. I know the 1000minds platform will enable for you to access your data rather swiftly but it is unclear if the data is made accessible to other than the platform license holder in this case. While the response of each individual to every choice task can be made available, the PAPRIKA algorithm is required to create the results from the current study. 

Response: We have already provided a datafile that shows the post-PAPRIKA criteria weights but we can add an additional datafile that shows the raw choice-task responses. It is possible for these data to be analysed using different DCE methods, however the selection and order of choice tasks that were presented in the DCE were created by the PAPRIKA algorithm within the 1000minds environment. Changes made t o the manuscript: An additional datafile has been uploaded to figshare. See page 2 line 15

---

## [Decision Letter · Decision Letter 1]

21 Nov 2023

Embedding stakeholder preferences in setting priorities for health research: using a discrete choice experiment to develop a multi-criteria tool for evaluating research proposals

PONE-D-22-25100R1

Dear Dr. Taylor,

We’re pleased to inform you that your manuscript has been judged scientifically suitable for publication and will be formally accepted for publication once it meets all outstanding technical requirements.

Kind regards,

Kekeli Kodjo Adanu, MB CHB, MPH

Academic Editor

PLOS ONE

Additional Editor Comments (optional):

Congratulations on the acceptance of your paper. Hopefully our readers and funding organizations will find it useful in the assessment of research studies.

Reviewers' comments:

Reviewer's Responses to Questions

**Comments to the Author**

1. If the authors have adequately addressed your comments raised in a previous round of review and you feel that this manuscript is now acceptable for publication, you may indicate that here to bypass the “Comments to the Author” section, enter your conflict of interest statement in the “Confidential to Editor” section, and submit your "Accept" recommendation.

Reviewer #1: All comments have been addressed

2. Is the manuscript technically sound, and do the data support the conclusions?

Reviewer #1: Yes

3. Has the statistical analysis been performed appropriately and rigorously? 

Reviewer #1: Yes

4. Have the authors made all data underlying the findings in their manuscript fully available?

Reviewer #1: Yes

5. Is the manuscript presented in an intelligible fashion and written in standard English?

Reviewer #1: Yes

6. Review Comments to the Author

Reviewer #1: Thank you for addressing my questions. No further comments. But I now can't submit unless I add extra characters to reach a minimum of 100....

7. PLOS authors have the option to publish the peer review history of their article (what does this mean?). If published, this will include your full peer review and any attached files.

Reviewer #1: No

---

## [Editor Report · Acceptance letter]

22 Nov 2023

PONE-D-22-25100R1 

Embedding stakeholder preferences in setting priorities for health research: using a discrete choice experiment to develop a multi-criteria tool for evaluating research proposals 

Dear Dr. Taylor:

I'm pleased to inform you that your manuscript has been deemed suitable for publication in PLOS ONE. Congratulations! Your manuscript is now with our production department. 

Kind regards, 

on behalf of

Dr. Kekeli Kodjo Adanu 

Academic Editor

PLOS ONE